# Research Progress on the Measurement Methods and Clinical Significance of Capillary Refill Time

**DOI:** 10.3390/s24247941

**Published:** 2024-12-12

**Authors:** Yuxiang Xia, Zhe Guo, Xinrui Wang, Ziyi Wang, Xuesong Wang, Zhong Wang

**Affiliations:** 1School of Clinical Medicine, Tsinghua University, 30 Shuangqing Road, Haidian District Beijing, Beijing 102218, China; chenjiu0525@163.com (Y.X.); xr-wang23@mails.tsinghua.edu.cn (X.W.); 15700060486@163.com (Z.W.); 2Beijing Tsinghua Changgung Hospital Affiliated to Tsinghua University, 168 Litang Road, Changping District, Beijing 102218, China; gza01482@btch.edu.cn

**Keywords:** capillary refill time, microcirculation, CRT measurement methods, sepsis, non-invasive monitoring, peripheral circulation

## Abstract

The monitoring of peripheral circulation, as indicated by the capillary refill time, is a sensitive and accurate method of assessing the microcirculatory status of the body. It is a widely used tool for the evaluation of critically ill patients, the guidance of therapeutic interventions, and the assessment of prognosis. In recent years, there has been a growing emphasis on microcirculation monitoring which has led to an increased focus on capillary refill time. The International Sepsis Guidelines, the American Academy of Pediatrics, the World Health Organization, and the American Heart Association all recommend its inclusion in the evaluation of the system in question. Furthermore, the methodology for its measurement has evolved from a traditional manual approach to semiautomatic and fully automatic techniques. This article presents a comprehensive overview of the current research on the measurement of capillary refill time, with a particular focus on its clinical significance. The aim is to provide a valuable reference for clinicians and researchers and further advance the development and application of microcirculation monitoring technology.

## 1. Introduction

Clinical researchers are now focusing on the problem of macrocirculation and microcirculation decoupling in the treatment of severe hemodynamics. Since the normalization of main circulation parameters has not resulted in the synchronous normalization of microcirculation perfusion and cell oxygen delivery, improving microcirculation has become one of the most important aspects of clinical treatment for many critically ill patients. Directly monitoring the microcirculation condition of deep organs in a clinical context is difficult, though. It is commonly used to evaluate the microcirculation status of the entire body, since the peripheral circulation status is easily observable and can represent the microcirculation status of the entire body. Therefore, the study and creation of peripheral circulation monitoring markers is clinically relevant.

Capillary refill time (CRT) represents the earliest indicator employed for the monitoring of peripheral circulation. It pertains to the time required for the blood to refill the capillaries following the release of pressure and the subsequent squeezing out of the skin [1,2]. A disruption in the peripheral circulation will result in a notable prolongation of the capillary refill time. As a crucial indicator for assessing microcirculatory function, it can directly reflect the perfusion of the microcirculation in the body, providing an essential foundation for clinical diagnosis and treatment [3,4]. However, the issue of the reliability and accuracy of this indicator’s measurement has yet to be fully resolved. The traditional measurement process is susceptible to a number of influencing factors, poor repeatability, and obvious observation errors, all of which have led to considerable debate regarding its sensitivity and specificity [5,6,7]. This, in turn, has limited its clinical application. In order to obtain accurate and reliable quantitative data, a great deal of in-depth research and practical exploration have been conducted in the clinical field in recent years, resulting in the development of a series of measurement tools. From the perspective of combining medicine and engineering, this article presents a comprehensive overview of the development and evolution of CRT measurement tools, outlining the advantages and disadvantages of each tool. It then provides an analysis of the research progress made in terms of its clinical significance and clinical application, with a view to offering new insights into the field of non-invasive monitoring of microcirculation.

## 2. Materials and Methods

### 2.1. Inclusion and Exclusion Criteria

The inclusion criteria for studies were as follows: (1) all studies were required to involve capillary refill time and (2) the full text must be written in English. The following studies were excluded from the review: (1) studies including literature with incomplete and inconsistent data and (2) repeated publications or studies for which the full text was not available.

### 2.2. Literature Search Strategy

A comprehensive search of the literature on CRT was conducted using a variety of databases, including PubMed, Web of Science, Google Scholar, Sci-Hub, The Cochrane Library, Embase, Scopus, and ScienceDirect. The search period was extended to 1 July 2024. A search was conducted using both subject words and free words, and relevant references were identified. The following search terms were employed: “capillary refill time”, “microcirculation”, “CRT measurement methods”, “sepsis”, “septic shock”, “non-invasive monitoring”, “peripheral circulation”, and “intensive care medicine”.

### 2.3. Literature Screening

The relevance of the titles and abstracts of all remaining studies was evaluated by two independent researchers. Studies that were deemed to be beyond the scope of the review were excluded. Any discrepancies were resolved through discussion or by a third researcher. Subsequently, the full text of the remaining studies was subjected to an eligibility assessment. If a study was deemed to be ineligible for inclusion, it was excluded from subsequent analysis. The reference list of all studies that underwent full-text screening was manually searched to identify any studies that were not retrieved during the initial search, a process which is referred to as a “reverse search”. Furthermore, all articles citing the full-text screened studies were also searched (i.e., a “forward search”). In the instance of studies potentially meeting the inclusion criteria being identified, they were evaluated in the same manner as previously described.

## 3. Results

### 3.1. Results of Literature Search

Following an initial search, 789 articles were retrieved. An additional 105 articles were obtained through citation tracing. Following a comprehensive and meticulous screening process, a total of 138 articles were ultimately deemed eligible for inclusion.

### 3.2. Normal Range of CRT

At the present time, manual measurement remains the most commonly used and most widely employed method. This method is characterized by significant inter-individual variability and a lack of standardized operating procedures. The reliability and accuracy of the measurement data have long been a matter of contention. Furthermore, the CRT is susceptible to external influences, including age, temperature, light, compression site, pressure, and time [8,9]. The definition of the normal range for the CRT remains a topic of contention. In 1981, experts in the field of trauma scoring reached a consensus and established the upper limit of normal CRT time as 2 s or less [10]. This upper limit value is widely known and utilized within the medical community. Nevertheless, with the ongoing advancement of hemodynamic research, an increasing number of researchers have begun to challenge the notion of establishing a unified CRT critical point for all age groups and clinical conditions. In 1988, the University of California Emergency Medicine Center conducted an empirical study to ascertain the normal value of CRT. The research team conducted a study to measure the CRT values of 100 healthy children, 104 adults, and 100 elderly volunteers. The results demonstrated that the CRT for adult women was 2.9 s, while that for older adults was 4.5 s [11]. The findings of this research provide a valuable reference point and scientific foundation for related fields of study.

Currently, CRT is a widely utilized method for the assessment of the hemodynamic status of newborns and children. It has been demonstrated that the normal range for this parameter in healthy newborns and children is considerable. Measurement sites include the head, chest, waist, palm, heel, instep, and so forth. There is considerable variation in the CRT of different sites [12]. The CRT of newborns is not affected by gender, the time of pregnancy termination, weight, gestational age, size of infant containers, or phototherapy. The upper limit of the CRT in newborns is 5–7 s, which is related to their immature skin. The upper limit of the normal value for the feet and chest of healthy children is 4.05 s (95% CI 3.61–4.49 s) [13]. There is an unexplained degree of heterogeneity in the measurement data obtained from other sites. Subsequently, in adulthood, the CRT increases at a rate of 3.3% per decade [14]. It may, therefore, be more scientific and accurate to define the range of CRT according to age and measurement site. Table 1 provides a summary of the values of CRT in different age groups and different sites for reference purposes. It is noteworthy that, while researchers may have differing opinions regarding the range of CRT in healthy individuals, the change in CRT as a monitoring value of peripheral perfusion is of greater importance than the baseline value [15]. The presence of the sympathetic–endocrine mechanism means that, when circulatory dysfunction occurs in the body, peripheral circulation is always sacrificed first in order to ensure perfusion of vital organs, and CRT increases at this time. As a result of the body’s intrinsic compensation mechanisms, fluid resuscitation, or the administration of vasoactive drugs, the microcirculation will gradually return to its normal state. In this process, restoration of the large circulation occurs first, followed by that of the peripheral circulation, which is restored last. This is accompanied by a decrease in CRT. Consequently, monitoring alterations in CRT is beneficial for the prompt identification of circulatory failure and for the real-time assessment of the effectiveness of fluid resuscitation. At present, research is underway within the medical community on the pharmacokinetics of CRT [16,17,18,19].

### 3.3. Measurement Method of CRT

#### 3.3.1. Manual Measurement

The methodology employed for the measurement of CRT, as initially documented in the literature, was notably vague and approximately equivalent to the definition provided, namely: the measurer applies pressure to the patient’s peripheral skin, causing it to turn white, and then records the time required for the skin color to recover after the pressure is removed. In clinical applications and clinical research, the selection of the measurement sites and measurement methods in different scenarios is more diverse. The measurement sites include the nail bed, palm, forehead, earlobe, chest, lower abdomen, knee, heel, sole, and so forth [27]. The pressure is typically applied with the objective of making the skin pale, and there is no clear pressure value. The pressing time varies from one to fifteen seconds. Manual measurement is the most prevalent CRT measurement method and is particularly well-suited to regions with constrained medical resources [28].

Furthermore, in 2019, the medical community began to adopt the microscope slide method as an alternative to the traditional manual pressing method. This involves applying firm and steady pressure with a slide, resulting in a noticeable change in the skin color, and using a precision timer to record the time required to restore the normal skin color [29]. Ait-Oufella H. has proposed a standardized measurement method for CRT as a result of a comparative study, whereby sufficient pressure is applied to the distal tip of the finger to create a thin white crescent-shaped pattern at the distal end of the fingernail. The patient is instructed to hold the pressure for 15 s, after which the pressure is released and the time required for the skin to regain its normal color is recorded. A precision timer is also used to take the average of two measurements [30]. Saito D. et al. have proposed that the standard posture for measuring CRT is the supine position with the hand at heart level, based on the findings of their research [31]. Lima A. has proposed the use of a stopwatch during the measurement process, with two measurements taken at one-minute intervals and the average of the two results calculated [32]. In addition to standardizing patient position and compression time, other researchers have also standardized the compression force using syringes [33,34]. However, the potential confounding factors of gender and age of the subjects, the observer bias of the tester, and the light intensity and temperature of the test environment may hinder the widespread use of CRT [35].

#### 3.3.2. Semi-Automatic Measurement Technology

In order to reduce or eliminate the influence of subjective factors in traditional CRT measurement methods and obtain quantitative data as an important basis for clinical diagnosis, Professor Shavit first proposed the concept of digitally measured capillary refill time (DCRT) based on the digital camera technology in 2006. The specific measurement method of DCRT is as follows: the subject is positioned in a supine manner, with a digital camera fixed at a distance of 3–5 cm from the fingertips. The tester applies pressure to the fingertips with a smooth rod for a period of five seconds before releasing it. The software employed for analysis is used to examine the alterations in the coloration of the patient’s fingertips in a series of frames. The time interval between the frame in which the pressure is removed and the frame in which recovery is observed is calculated and recorded as DCRT [36]. Nevertheless, the routine implementation of this apparatus in clinical contexts is not a viable proposition. Video data recorded in a complex clinical setting may be prone to instability and it may be challenging to achieve focus.

One of the pivotal issues in CRT measurement is the discrepancy between observers [37,38]. The accumulation of personal work experience and the undertaking of professional training may facilitate an improvement in the accuracy of CRT measurement [18]. In 2019, Chiba University in Japan proposed the development of a wearable device based on visual feedback technology (Figure 1) [39]. The device’s key advantage is that it quantifies the intensity and duration of the pressing action through visual feedback, thereby avoiding the potential influence of pressure and pressing time on the measurement results. In the same year, Kawaguchi R. et al. developed a new device with an adjustable pressing force and time. The results of the experiment demonstrated that measurements of pressing time below two seconds were unreliable. It is, therefore, recommended that measurements of CRT be conducted using a pressure of between three and seven Newtons applied to the peripheral skin for a period of two seconds. The findings of this study also offer guidance to researchers on the implementation of standardized clinical procedures [40]. It is pertinent to highlight that these methodologies may not be optimal for individuals with darker skin tones or in environments with low illumination.

Given that the reflected light undergoes a change in intensity when pressure is applied to the skin, and that the photoplethysmogram of the reflected light is reflective of alterations in blood volume, numerous scholars have put forth novel methodologies for quantifying CRT based on this phenomenon [41,42,43]. Additionally, it is important to note that the duration and intensity of applied pressure can also affect the measured CRT. In order to address this issue, Chong Liu and colleagues combined a pressure sensor with a photoelectric volume pulse wave sensor and employed the exponential regression method to fit the real measured light intensity (Figure 2). This allowed them to simultaneously monitor the blood oxygen saturation and pressure of the subjects [44,45]. In 2021, Ballaji HK and colleagues used this integrated fiber optic sensor to detect changes in plantar skin blood volume and contact pressure. This resulted in the acquisition of plantar skin CRT [46]. The sensor in question is compact yet highly capable, with the potential to significantly enhance the precision and efficiency of CRT measurement. However, there is currently a paucity of clinical studies exploring its applications.

#### 3.3.3. Fully Automatic Measurement Technology

Presently, the majority of fully automatic measurement devices are based on pressure sensing technology and photoplethysmography technology. The distinction can be attributed to the design of the pressing actuator. The most prevalent approaches include the utilization of a pneumatic pressure application system and a mechanical pressing actuator. In 2016, Blaxter et al. published details of a pneumatic CRT measuring instrument. The pneumatic pressure device within the measuring instrument is responsible for the application and release of the pressure. The airbag in the measuring instrument exerts pressure on the skin, resulting in its whitening. Subsequently, a multiwavelength reflection measurement device is employed to ascertain the diffuse reflectance value of the skin with precision (Figure 3 and Figure 4) [47]. The device eliminates the observer bias by standardizing the application and release of pressure and by electronically measuring the diffuse reflectance, thereby achieving the objective of continuous and repeated measurement of CRT and reducing the influence of human factors on the results. Subsequently, in 2020, Japanese and American research teams announced the development of a pneumatic fingertip compression device that employs pneumatic pressure to measure the CRT of the index fingertip [48]. In comparison to electromechanical pressurization, these pneumatic pressure systems (such as blood pressure cuffs and pneumatic respiratory sensors, both of which are widely used in clinical practice) are more cost-effective. However, since not all pressure can be released instantaneously, the resulting measurement is biased. The mechanic-based press execution device [49,50,51] can control the pressure element to press, maintain pressure, and return to the normal pressure of the measuring finger according to a pre-programmed computerized sequence, effectively solving the problem of inconsistent position and pressure caused by the manual pressing of capillaries in the traditional measurement process and further mechanizing and standardizing peripheral circulation monitoring.

These fully automatic measuring devices employ photoelectric sensors to detect blood oxygen signals in real-time, thereby facilitating the real-time monitoring of oxygen saturation levels in the blood. The aforementioned signals are in the form of discrete digital ones. Subsequent processing of the signal enables the acquisition of several crucial parameters, including capillary refill time. By calculating the first derivative of the measured value, Morimura [52] obtained Q-CRT, a time parameter from the press release point to a refill of 90% of the original blood volume. This parameter was demonstrated to be related to tissue hyperperfusion. Furthermore, the results may be expressed using the method of photoplethysmography (PPG). In 1933, Hertzman et al. [53] were the first to identify a correlation between infrared light and blood volume. The advancement of hemodynamics and the growing popularity of this non-invasive detection method have contributed to the widespread adoption of PPG as a detection means in recent years. Previously, the analysis of PPG signals was primarily conducted through the examination of various parameters derived from the first and second derivatives. This approach allows for the identification of each key stage of PPG signal change, thus enabling the determination of physiological information or potential diseases. In recent years, the development of artificial intelligence and deep learning technologies has led to an increasing number of studies demonstrating that the utilization of deep learning technologies can achieve superior outcomes and circumvent the necessity of introducing an excessive number of assumptions during the data processing phase. This is attributed to the capacity of neural networks to recognize features. For example, Voisin [54] posited that real-time monitoring of cardiac arrhythmias via photoelectric sensors integrated into smart wristbands is a crucial step in the early detection and treatment of cardiovascular diseases. It is of particular importance to achieve an accurate and scientific measurement of the corrected QT (QTc) interval and develop an automated analysis and processing system for the data. To facilitate the greater utilization and promotion of mechanized and automated testing methods, it is essential to engineer the entire process, from the hardware to the software, in a scientific and reasonable manner.

### 3.4. Clinical Significance of CRT

The clinical application of CRT can be traced back to 1910, when Takayesu and Lozner initially documented its use as an indicator of dehydration [55]. Subsequently, CRT has been progressively accepted and acknowledged by the medical community, becoming a pivotal diagnostic instrument for assessing dehydration. Over time, the range of CRT applications has gradually increased. In particular, in 1940, Guedel introduced it to the field of surgery and employed it to monitor the state of shock during surgical procedures by measuring the CRT of the forehead [56]. In 1947, Crismon and Fuhrman conducted further research on the applications of CRT. They discovered that CRT could be employed as an indicator of intraoperative shock [57]. Subsequently, CRT has been progressively adopted as a straightforward method for evaluating shock and dehydration. In 1981, Champion first proposed the integration of CRT into the digital trauma score, advocating its use as a diagnostic indicator in shock assessments [10]. As a non-invasive, straightforward, simple, and repeatable method of assessing peripheral microcirculatory perfusion [58], CRT was later incorporated into the American Advanced Trauma Life Support Manual [59].

As an effective assessment indicator of the hemodynamic status of newborns and children, CRT has been included in numerous international guidelines, including the Advanced Paediatric Life Support (APLS), ACCM Guidelines for Septic Shock in Children and Neonates, Guidelines for the Management of Septic Shock and Septic-Related Organ Dysfunction in Children, and NICE Guidelines for Febrile Illness in Children [60,61,62,63,64]. It has been employed in the context of pediatric advanced life support for over four decades [65]. The combination of CRT with other clinical signs (such as tachycardia and dry mucous membranes) can facilitate the diagnosis of dehydration in children. Studies have demonstrated that CRT exhibits a high degree of specificity, ranging from 88% to 94%, in identifying children with moderate dehydration (≥5%) [66]. In the field of pediatrics, the most compelling piece of evidence supporting the use of CRT is its capacity to predict the mortality of critically ill children. Prolonged capillary refill time (PCRT) is an indicator of poor hemodynamic status in patients with sepsis. It is a clinical tool that is associated with disease severity and tissue perfusion insufficiency and is a reliable indicator of mortality in critically ill patients [67,68]. Results from such research indicate that children with prolonged capillary refill time (PCRT) exhibit a fourfold increase in the risk of mortality compared to those with normal CRT. Furthermore, the findings suggest that PCRT could serve as a predictor of death from malaria, hypoxia, malnutrition, dengue fever, sepsis, and meningitis. The measurement of CRT has been incorporated into the routine protocols of the integrated management of childhood illnesses (IMCI) program [61]. The diagnostic specificity of PCRT is high, while its sensitivity is low [66]. Consequently, PCRT can be employed as an indicator of potential severity, facilitating the triage of children. It is imperative that children with PCRT receive prompt treatment; however, the absence of PCRT does not entirely eliminate the possibility of a serious underlying illness.

The use of CRT is beneficial for the early identification of patients who are critically ill. The body’s intrinsic neuroendocrine mechanism initiates the perfusion of the skin tissue that is sacrificed in the initial phase of a circulatory failure, thereby ensuring the blood supply to vital organs such as the heart, lungs, and brain. A substantial body of evidence from numerous studies has demonstrated that alterations to peripheral circulation perfusion parameters, such as CRT and center-to-extremity temperature difference, in patients with severe illness and injury are not aligned with those observed in major circulation parameters. It is not uncommon for peripheral circulation dysperfusion to manifest prior to the emergence of abnormalities in major circulation parameters. Consequently, CRT, as a direct evaluation index of peripheral circulation perfusion, should prove beneficial for the early identification of severe patients. Furthermore, CRT represents the most direct approach to monitor the blood flow to the distal capillaries, and a prolonged CRT indicates impaired capillary perfusion. In reality, the perfusion status of the capillaries is dependent on a series of complex factors, including capillary perfusion pressure, arteriole tension, the density of open capillaries, and the function of blood cells. The state of the central circulation and the function of the local aorta are the determining factors in capillary perfusion pressure. The tension of small arteries and the opening of capillaries can be affected by both internal and external vasoactive substances. Additionally, changes in the function of blood cells can also result in the stagnation of blood flow in capillaries. Consequently, an extended CRT can be observed in association with a number of pathological conditions, including shock of various etiologies, obstructive lesions of the limb arteries, the administration of vasoconstricting drugs, frostbite, vasculitis, and others.

Prolonged CRT is a significant predictor of patient prognosis. The extant evidence suggests that CRT measurement is most effective in assessing patients in shock. It is an effective method for assessing the severity of the disease [69,70,71,72,73,74,75,76,77,78] and guiding the treatment of patients in shock [79,80,81,82,83]. From a cellular perspective, the fundamental issue underlying shock is an inadequate oxygen supply to the cells, impairing the maintenance of essential cellular functions and, ultimately, leading to organ damage. In the initial phase of shock, there is a redistribution of the blood from non-vital peripheral organs to central organs. Consequently, inadequate perfusion of the peripheral organs occurs even before alterations to macrocirculatory hemodynamic variables [84,85]. One of the primary objectives of hemodynamic monitoring is to identify instances of early perfusion and oxygenation insufficiency. The timely administration of treatment and guidance for resuscitation, based on the monitoring results, can prevent organ damage. Patients with prolonged CRT have been shown to exhibit a markedly elevated risk of mortality. It has been demonstrated that the accuracy of quantitative CRT in predicting sepsis is comparable to that of the qSOFA score and blood lactate level [86]. It is an independent predictor of mortality at 28 days in critically ill patients [87]. The early monitoring of CRT has been demonstrated to have a beneficial impact on mortality and prognosis in patients with sepsis [88,89]. It is an ultrasound substitute for vascular tension of visceral organs in the early stages of shock [4].

CRT can be employed to guide patient treatment. For patients presenting with shock, macrocirculatory stabilization represents merely the initial stage; improvement in microcirculation is the crucial objective [90,91,92]. As a flow-sensitive indicator, the normalization of CRT is parallel to the increase in local blood flow [93,94]. It is, therefore, evident that CRT monitoring is of significant importance in the context of fluid resuscitation [95]. In comparison to other indicators, such as urine output and lactate, CRT may be a more sensitive, rapid, and effective method for guiding fluid resuscitation. A randomized trial published in the Journal of the American Medical Association (JAMA) has demonstrated that a resuscitation strategy targeting CRT normalization may result in a reduction in the morbidity and mortality of patients with septic shock in comparison to a strategy based on lactate clearance [85]. Furthermore, a recent prospective multicenter observational study has demonstrated that CRT is associated with the early prediction of 90-day mortality and venoarterial extracorporeal membrane oxygenation (VA-ECMO) support requirements in patients suffering from cardiogenic shock [96]. In light of these findings, international experts have recommended CRT as a potential recovery target [97]. Furthermore, in critically ill patients, vasoconstriction and reduced tissue perfusion may occur when increasing cardiac output and arterial pressure with vasoconstrictor drugs. The evaluation of peripheral circulation can inform the use of vasoconstrictor drugs from a variety of perspectives. For instance, in the event of a significant deterioration in peripheral circulation following the administration of high doses of norepinephrine, a reassessment of the situation may be warranted. It is necessary to adjust the medication in an appropriate manner.

These studies have corroborated the hypothesis that CRT, as a potential resuscitation target, is no less clinically efficacious than existing resuscitation targets. Furthermore, it has been demonstrated that CRT is a valuable supplement and enhancement to existing resuscitation indicators. While there may be certain issues associated with the use of CRT as a standalone method for identifying critically ill patients, when CRT is used in conjunction with other peripheral circulation perfusion indicators, its reliability is significantly enhanced. At present, the International Sepsis Guidelines, the American Academy of Pediatrics, the World Health Organization, the European Resuscitation Council, and the American Heart Association all advocate for the incorporation of CRT measurements into the evaluation of healthcare systems [59,98,99,100,101]. CRT monitoring is of paramount importance in ensuring that patients receive prompt and efficacious treatment, and its stability and reliability provide substantial support for clinical decision-making.

## 4. Discussion

Both freehand compression and slide-based measurements are fast, convenient, noninvasive, and inexpensive methods of assessing a patient’s hemodynamic status and are widely used in clinical practice. However, this overly subjective measurement method has led to many questions about the accuracy and reliability of its results. Researchers have invented some semi-automated measurement devices to improve the repeatability and accuracy of CRT measurements, such as capturing skin color changes using video cameras and ensuring blood emptying by measuring contact pressure, etc. These devices have improved the accuracy of the measurements to a certain extent, but they have not been put into clinical use due to their cumbersome operation and the existence of human interference factors. Fully automated measurement devices based on pressure-sensing technology and optoelectronic volumetric pulse wave technology can accurately and efficiently determine a patient’s hemodynamic status, with good instrument repeatability and elimination of human interference. Pneumatic pressure application systems are inexpensive but can lead to biased measurements due to their inability to release the full pressure instantaneously. Of all the measurement modalities, fully automated measurement devices based on mechanical compression provide the most accurate measurements and good instrument repeatability. The advantages and disadvantages of the different measurement methods are summarized in Table 2.

Developing these digital and automated CRT measuring devices may improve the accuracy of hemodynamic monitoring, but researchers have encountered many difficulties in environments/settings with limited resources. For example, in recent years, with the increasing number of pieces of monitoring, treatment, and first-aid equipment in intensive care units and emergency departments, the available space has been shrinking, and the large-volume and single-output instrument is difficult to be accepted clinically. After the shelving of a small range of clinical trials, it is difficult to integrate an instrument into clinical practice. On the one hand, it is recommended that devices that can obtain other biometric indicators at the same time as CRT measurements are developed. For example, a temperature sensor that measures ambient temperature or skin temperature can be embedded in the sensor that measures the CRT to provide a temperature correction to avoid the impact of skin and ambient temperature on the CRT. In addition, it can also be combined with key clinical parameters such as heart rate and pulse oxygen saturation to build a comprehensive and accurate diagnostic tool to better assist clinical work. On the other hand, researchers also need to consider the cost of the equipment and its portability, with follow-up research suggesting that the current high cost of the measuring instrument and its large desktop can be upgraded to a handheld measuring instrument, simplifying the measurement process and reducing the workload of medical personnel.

## 5. Conclusions

As a principal indicator of the dual effects of compensatory mechanisms and vasoactive drugs in critically ill patients, CRT is the first to be compensated and the last to recover. This makes it an important basis for the early detection of acute circulatory failure and for the evaluation of the efficacy of fluid resuscitation in shock patients. Indeed, the perfusion state of capillaries is dependent on a number of intricately linked factors, including capillary perfusion pressure, arteriolar tension, open capillary density, and the functionality of blood cells, among others. The state of the macrocirculation and the function of local large arteries are the determining factors of capillary perfusion pressure. Both endogenous and exogenous vasoactive substances have the potential to influence arteriolar tension and capillary opening. Additionally, alterations to the functionality of blood cells may result in blood stasis within the capillaries. Consequently, PCRT can be documented in association with a number of pathological conditions, including shock resulting from various causes, limb arterial obstructive lesions, the utilization of vasoconstrictive pharmaceuticals, frostbite, vasculitis, and so forth. Presently, the clinical application of and the related research on CRT are primarily focused on the fields of neonatology, pediatrics, emergency medicine, and intensive care. The accurate monitoring of CRT is of great significance for patients with sepsis, shock, and dehydration. It is noteworthy that microcirculatory disorders represent a critical pathological basis for organ damage in patients suffering from various chronic metabolic diseases, thereby exacerbating the severity of the disease [102]. Abnormalities in microcirculation may manifest prior to the onset of the symptoms, serving as an early indicator of chronic metabolic disease. As a direct indicator of the body’s microcirculatory perfusion, CRT will undoubtedly play an important role in the early detection, prevention, and treatment of these diseases, and it is expected to bring a new perspective to the diagnosis and treatment of chronic metabolic diseases.

Due to time constraints, this paper only reviewed research on CRT measuring tools in the databases considered, and subsequent research can expand the search scope by, for example, including related invention patents, news reports, and other content. In addition, it must be pointed out that, although some progress has been made in the research and development of mechanical fully automatic measuring equipment, as of now, this type of equipment has not yet achieved large-scale clinical application. On the one hand, there is no unified standard for pressing time and force, both of which depend on the in-depth research on the judgment mechanism of individualized capillary closure time in the medical field. On the other hand, the engineering field also needs to fully consider that the pressing actuator must have the ability to release pressure instantly and bear the characteristics of being light and easy to carry. Therefore, engineers need to develop accurate, efficient, and compact measuring equipment.I In order to promote the widespread application of mechanical fully automatic measuring equipment in clinical practice, it must urgently be released to medical experts and engineering experts. The medical team should focus on the practicality and safety of CRT measurements and conduct in-depth assessments and exploration of the compression time and force control, clarifying the mechanism of individualized capillary closure time determination and setting the control logic based on clinical scenarios. The engineering team, on the other hand, should focus more on science and innovation, as well as on the design of the compression actuator and the standardization of the PPG signal. After completing the samples to the maximum satisfaction of the medical team, the medical team should then conduct several rounds of clinical trials to continuously improve the performance and reliability of the device and provide a more accurate and efficient measurement tool for clinical diagnosis and treatment.

## Figures and Tables

**Figure 1 sensors-24-07941-f001:**
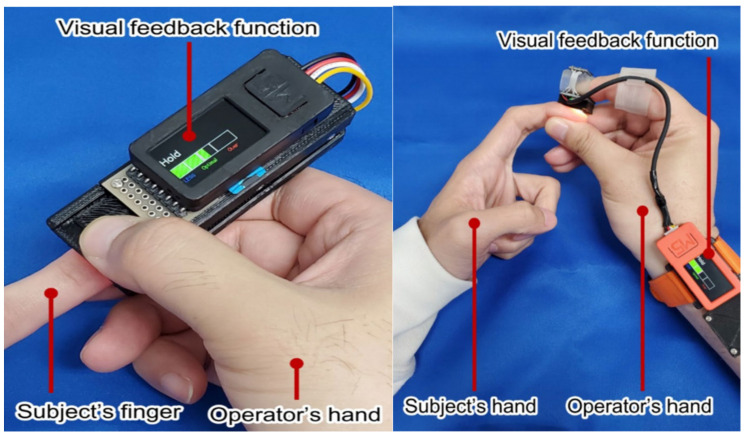
Wearable CRT measuring instrument based on visual feedback technology [39].

**Figure 2 sensors-24-07941-f002:**
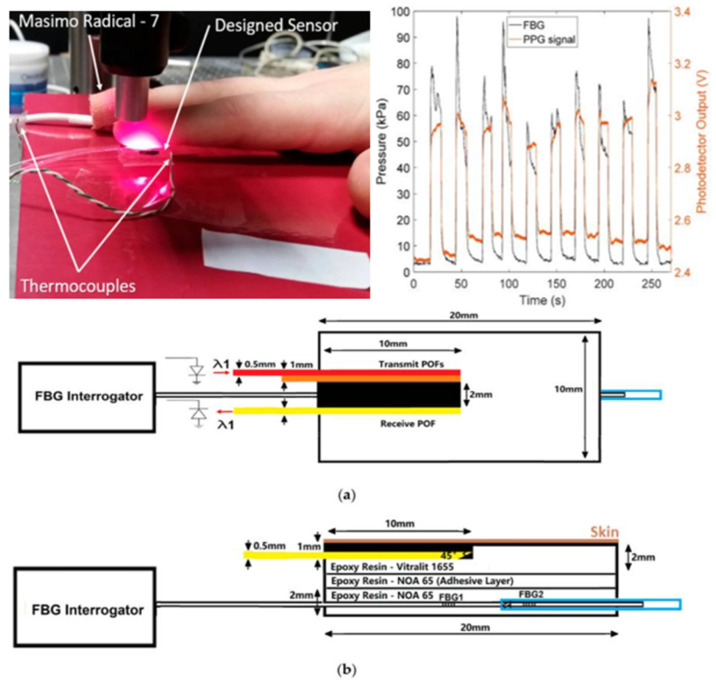
CRT measuring instrument test machine based on pressure sensing technology and photoelectric capacitance wave technology: (**a**) schematic diagram of a plane view of the probe. The red frame represents the POF connected to an infrared LED (center wavelength λ = 850 nm). The yellow frame represents the POF connected to a photodiode. As the sensor had been adapted from a pulse oximeter described previously, there was an additional channel (orange frame) that existed which was not used in this investigation. The blue frame represents the stainless steel tube. (**b**) Schematic diagram of a side view of the probe. The two sensors were encased in epoxy. The POFs transferred light from/to the skin, whilst the FBGs monitored the applied pressure. POF—polymer optical fiber; FBG—fiber Bragg grating; LED—light emitting diode [44].

**Figure 3 sensors-24-07941-f003:**
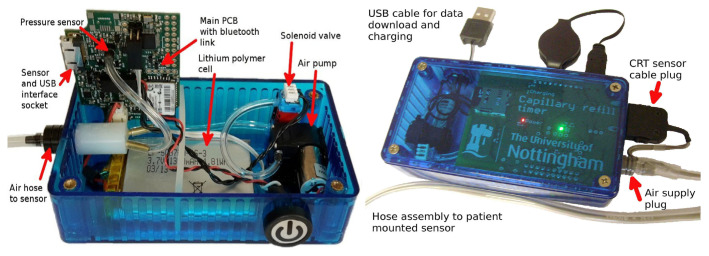
Wearable CRT measuring instrument based on visual feedback technology [47].

**Figure 4 sensors-24-07941-f004:**
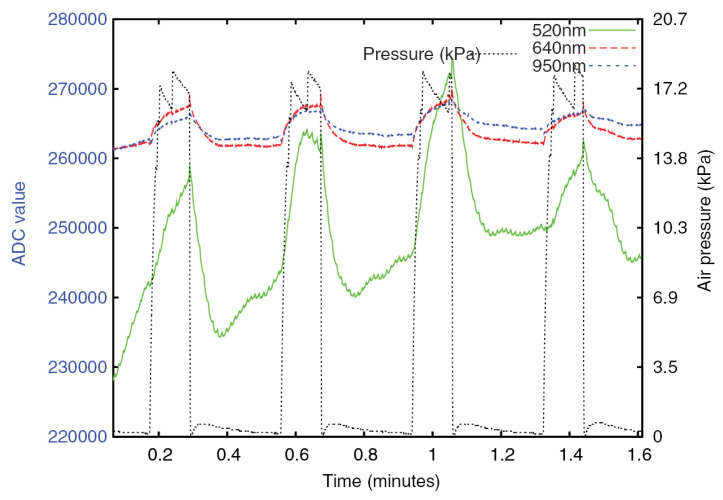
Actual test image of the pneumatic CRT measuring instrument test machine: typical raw optical intensity data from the base unit, prior to any processing [47].

**Table 1 sensors-24-07941-t001:** Normal capillary refill time range according to age and location.

Age Range	Sample Size	Type	Measurement Methods	Upper Limit of CRT	Refs.
Early neonatal period	42	Observational, cross-sectional	Use the fingers, soles of the heels, and sternum to press for 3–4 s, using a digital stopwatch.	Fingers: 2 s, chest, feet: 4 s	[20]
Newborn	627	Observational	Press the middle of the sternum for 5 s (supine position), take a measurement three times, and calculate the average.	3 s	[21]
	469	Cross-sectional	Press the chest, head, palms, and heels for 5 s, manually taking the time.	3 s	[22]
	137	Cross-sectional	Apply pressure to the right hand and the instep of the right foot for 5 s.	Hand: 4.23 ± 1.47 s, foot: 4.64 ± 1.41 s	[23]
Infants and children	92	Prospective, method comparison study	Apply enough pressure to the fingertips and sternum for 5 s to blanch the skin.	2–3 s	[24]
Teenager	20	Observational	Using a digital camera, the average of the first and fifth toes is used as the CRT.	3.5 s	[25]
Adult	1000	Prospective observational study	Use medium pressure to press the index finger of your right hand to the white point for 5 s and use a stopwatch to measure the time.	3.5 s	[26]
Elderly	1000	Prospective observational study	Same as above.	4.5 s	[26]

**Table 2 sensors-24-07941-t002:** Capillary refill time measurement methods.

Methods		Advantages	Disadvantages	Refs
1. Manual measurement	Hand pressing method.	Fast, convenient, non-invasive, and inexpensive.	Observer bias and poor reproducibility.	[28]
	Slide press method.	The color changes significantly, and the accuracy is higher than that of the bare hand pressing method.	Observer bias and poor reproducibility.	[29]
2. Semi-automatic measurement technology	Based on digital camera technology.	Accurate judgment of blood emptying and filling time points.	Data collected in a cumbersome, cluttered clinical environment are unstable and difficult to focus on; not suitable for people with darker skin or in poorly illuminated testing environments.	[36]
	Wearable devices based on visual feedback technology.	Quantification of compression intensity and time to avoid their influence on measurement results.	It requires manual pressing by the operator and is easily affected by external light.	[39]
	Based on pressure-sensing technology and photoelectric capacitance wave technology.	It ensures blood drainage by measuring the contact pressure.	CRT measurement for fingers only.	[40,41,42,43,44,45,46]
3. Fully automatic measurement technology	Based on diffuse reflection and pneumatic pressure application system.	It eliminates observer bias through the standardization of pressure application and release and through the electronic measurement of diffuse reflectance.	Unable to release all the pressure instantly, resulting in larger measurement results.	[47,48]
	Based on mechanical press actuators.	Accurate and efficient.	Single function, high cost, large instrument size, and inconvenient to carry.	[49,50,51]

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
