# Peer review of "Research Progress on the Measurement Methods and Clinical Significance of Capillary Refill Time"

_sensors, 2024, doi:10.3390/s24247941_

Round 1
Reviewer 1 Report
Comments and Suggestions for Authors
This article gives a concise summary of the therapeutic significance of CRT, its use in microcirculation monitoring, and the emphasis on combining engineering and medicine.
However, the following points need to be addressed:
1. The authors are suggested to avoid repetitive words eg: non-invasive, easy to implement, convenient, and too-long sentences throughout the text.
2. Explain in detail about the variation and its effects on clinical use, such as why using a consistent CRT threshold for all patient demographics is problematic.
3. The authors could shorten the material and method section and make it more reader-friendly.
4. The authors are suggested to mention the limitations of the study and the use of CRT in different populations.
5. The authors must consider adding further details about the obstacles in environments/settings with limited resources, even though the difficulties of implementing automated devices are discussed by the authors.
6. The authors should mention the clinical implications of CRT as a diagnostic indicator in detail.
7. The manuscript concludes strongly by highlighting the importance of interdisciplinary cooperation. The authors should emphasize explaining how overcoming these obstacles fits in with the more therapeutic goals that were previously covered.
Comments on the Quality of English LanguageKindly use short sentences and check for grammatical errors throughout the manuscript.
Reviewer 2 Report
Comments and Suggestions for Authors
A timely and useful review of developments in CRT monitoring and its applications. The paper should be of interest to the journal audience. O recommend addressing the following:
Line 19 - sensors is an international journal so it's not clear 'at home and abroad' is appropriate. Unless there are specific difference in the drivers for using CRT in China then this should be rephrased. Also line 130 discusses ‘foreign’.
Section 1 – although it’s not the main aim of the paper, a little more detail on the physiological origin of the CRT signal and why it is a useful representation of perfusion would help to set the scene better.
Line 37-38 – assessing shock and dehydration, it would be useful to discuss somewhere in the paper potential cross talk between these. As noted in Section 2, many factors/conditions affect CRT so signal interpretation in the presence of cross talk is important.
Line 42 - insert reference to American Advanced Trauma Life Support Manual.
Line 57, Section 2 – I don’t think materials and methods is the appropriate heading for this section. In this section, I recommend describing the search terms, databases and how you decided which papers to include, and why you decided to structure the paper in this way. A visual representation e.g. flowchart would be helpful. The Results section could then be the results of the review i.e. the papers found under these headings (currently described in section 2).
Line 59 – I’m not clear why both ‘most commonly used and most widely used’ is necessary. Just one will be sufficient.
Line 78 – ‘The CRT of newborns is not affected by gender, pregnancy …’ – I’m not sure what is meant by pregnancy in this list
Line 87 – ‘.. the change in CRT as a monitoring value of peripheral perfusion is more important than the baseline value[19].’ – given the range of absolute values this is a relevant point. Is there any indication of the change (or % change) in CRT that would be an indicator of a change in health status?
Line 100 – ‘and then observe the time it takes for the skin to return to the baseline color after 15 seconds’ – I’m not clear what this means, if one is just observing the time, what is the relevance of 15s?
Line 183 – Section 2.2.3 focuses on the instrumentation. I think it would be worthwhile briefly discussing the signal processing required to obtain CRT. Is there a standard change in intensity that is used e.g. time taken for the intensity to change from 90% to 10% of the maximum (or 100%-0%?). How is the intensity maximum obtained if the signal is not stable or there is noise present e.g. should the maximum be used, or an average at the plateau? Should curve fitting be used or the raw data? There are several ways to calculate CRT and a standardised process would be useful in future.
Line 219 – ‘..and so on.’ It’s not very clear what is meant by this, either explain what you are referring to or remove. I would generally avoid using etc in the paper unless it's obvious what this refers to.
Line 224-228 – the references [53-57] to the guidelines are very useful. However they are not current, the most recent is 2013, the oldest 2000. Can you check through the current guidance and update please.
Line 279 – again these guidelines are useful and more recent. Can you check through the current guidance and update if appropriate.
Table 2 – can you add a column for references and add the relevant reference numbers
Section 4 ‘Results’ would be better as ‘Conclusions’
The paper would benefit from proof reading as it contains many minor typos e.g. line ‘and that for older adults it was 4.5’ should be ‘and that for older adults was 4.5’.
Round 2
Reviewer 1 Report
Comments and Suggestions for Authors
The manuscript could be accepted in its current form.